# Moving toward a Greener China: Is China's National Park Pilot Program a Solution?

**Gonghan Sheng [1], Heyuan Chen [1], Kalifi Ferretti-Gallon [1], John L. Innes [1], Zhongjun Wang [1,2], Yujun Zhang [2] and Guangyu Wang [1,*]**

[1]  National Park Research Centre, Faculty of Forestry, University of British Columbia, Vancouver, BC V6T1Z4, Canada; jerrys92@mail.ubc.ca (G.S.); jacksonchy@alumni.ubc.ca (H.C.); kalifi.ferretti-gallon@ubc.ca (K.F.-G.); john.innes@ubc.ca (J.L.I.); wangzj814@bjfu.edu.cn (Z.W.)

[2]  National Park Research Lab, School of Landscape Architecture, Beijing Forestry University, 35 Qinghua E Rd, Beijing 100091, China; yjzhang@bjfu.edu.cn

\*   Correspondence: guangyu.wang@ubc.ca; Tel.: +1-604-822-2681

**Abstract:** National parks have been adopted for over a century to enhance the protection of valued natural landscapes in countries worldwide. For decades, China has emphasized the importance of economic growth over ecological health to the detriment of its protected areas. After decades of environmental degradation, dramatic loss of biodiversity, and increasing pressure from the public to improve and protect natural landscapes, China's central government recently proposed the establishment of a pilot national park system to address these issues. This study provides an overview of the development of selected conventional protected areas (CPAs) and the ten newly established pilot national parks (PNPs). A literature review was conducted to synthesize the significant findings from previous studies, and group workshops were conducted to integrate expert knowledge. A qualitative analysis was performed to evaluate the effectiveness of the pilot national park system. The results of this study reveal that the PNP system could be a potential solution to the two outstanding issues facing CPAs, namely the economic prioritization over social and ecological considerations that causes massive ecological degradation, and the conflicting, overlapping, and inconsistent administrative and institutional structures that result in serious inefficiencies and conflicts.

**Keywords:** national park; pilot program; environmental sustainability; governance; economic development

## 1. Introduction

National parks have played an important role worldwide in protecting natural landscapes [1–3]. The use of national parks as a significant tool for nature conservation and wilderness preservation has been adopted globally by most countries and utilized to achieve ecological protection and biodiversity restoration [4].

Despite the multiple policies to address ecological issues, biodiversity in China continues to decline [5]. Protected area management in China is hampered by a complicated hierarchy of management and inconsistent regulatory standards across various departments, resulting in a lack of effective communication and information sharing [6].

In 2008, the Chinese government took a step towards improving the quality of protected areas with the approval of China's first national park management office and introduced a pilot national park system in 2015. This initiative is regarded as an experimental application of an international model of national parks, with the expectation that these parks can better serve to protect biodiversity and promote human livelihood [7].

*History of National Parks in China*

As defined by the International Union for the Conservation of Nature (IUCN), a protected area can be considered a national park if it conforms to the following criteria: (1) has a large natural area, (2) is set aside to protect large-scale ecological processes, (3) supports endemic biodiversity, and (4) facilitates cultural, scientific, educational, recreational, and visitor opportunities [8]. Since the introduction of the world's first national park in 1872 (Yellowstone National Park), many countries have adopted a national park system to enhance the protection of their natural landscapes.

In China, the conceptualization of protected areas dates back to the Qing dynasty. For instance, forest protection and restoration were codified into Qing statutory laws with the institutionalization of official positions (appointed by the central authority) for management and supervision [9]. However, these protected areas were mainly private "gardens" belonging to nobles, temples, and cemeteries. Since the 13th century, the Bogd Khan region has been considered home to one of the holiest mountains in Mongolia and is still regarded as a pilgrimage site by Buddhists. In 1783, the Qing government listed Bogd Khan as a protected area due to the increasing demand for wild animals and plants used in religious activities. In 1996, the United Nations Educational, Scientific and Cultural Organization (UNESCO) officially recognized this as a protected area. In a sense, this is one of the oldest protected areas in the world. At the end of the Qing dynasty and the beginning of the Republic of China in 1912, the region was still being colonized by western countries, and its national strength was weak. At that time, managed natural landscapes were dominated by royal and private gardens (parks). Only some private gardens were open to the public for free. In the 1930s–1940s, the Chinese government tried to build a national park system and developed national parks associated with scenic spots, such as Lusha and Taihu. In 1930, Chen Zhi published the "National Taihu Park Plan", which was intended to better preserve the country's high-quality natural and cultural resources and to allow the public to enjoy its natural scenery. In 1936, after the government of the Republic of China nationalized ownership of Lushan Mountain, it proposed a "National Park Plan" for it, based on the original scenic sightseeing and summer vacation features of Lushan Mountain. This opened the way to the establishment of Chinese national parks. In 1939, with the outbreak of the Second World War and the Chinese Civil War, the construction of the parks was abandoned [10].

Since the formation of the People's Republic of China in 1949, China has been relatively slow in adopting the concept of national parks. While China's first National Nature Reserve was established in 1956 in the region surrounding Dinghu Mountain, a protected area with the expressed objective of nature conservation, Zhangjiajie National Forest Park was formally introduced in 1982, after China implemented its Reform and Opening-up Policy in 1978 [2,11]. Consequently, scholars noted a shift in public attitude and corresponding policies towards the importance of national parks in mitigating environmental degradation, followed by a proliferation in the number of protected areas in China. Currently, there are over 2700 protected areas under numerous designations (e.g., national nature reserves, national forest parks, national scenic areas, world heritage sites, world geoparks, and national marine protection areas) [12]. However, experts argue that many of these protected areas are ineffective in maintaining ecological integrity and are thus considered "Paper Parks", suggesting they exist in name only [4] (p. 248), [6,11]. One study estimates that, despite an increase in the number of nature reserves (a common protected area designation in China) between 2007 and 2014, their total area decreased by 3% [13]. Moreover, although China has included the words "national" and "park" in protected area titles (such as national forest parks, national nature reserves, and national geoparks), they are nominal and without the practical functions of an IUCN-classified national park. Furthermore, China's protected area management is administered by different government bodies, and therefore management is inconsistently applied throughout this network of conventional protected areas (henceforth CPAs). Experts have referred to CPA management as "fragmented" and messy, failing to prevent environmental degradation while mainly focusing on economic gains [4,6], [14] (p. 762).

The development of national parks in China has had two dominant driving forces: a growing middle class with an interest in nature-based tourism and recreation, as well as a growing public concern

over the depletion of natural resources [4]. These two trends are interdependent and correspond to the IUCN's primary recreational and ecological objectives of a national park. However, growing demand for outdoor recreation has encouraged CPA managers to prioritize revenue over social and ecological considerations [4,14], resulting in degradation of the country's local ecosystems [4,15,16]. Moreover, the administrative and institutional structure of the CPAs has led to conflicting, overlapping, and inconsistent mandates due to the long-lasting involvement of multiple ministries [17–20]. In 2013, China's Central Committee proposed the "Establishment of a Pilot National Park System" as a top-down design to enhance protected area management for ecological prioritization by addressing the long-standing administrative and institutional issues [17–19]. In 2017, after further amendments, the park proposal was formally enacted through the Development of a National Park System Overall Plan, which aimed to complete the establishment of ten PNPs by 2020 [19,21,22].

The overall goal of this paper is to evaluate the ability of the national park model to resolve two outstanding problems of the CPAs—the economic prioritization over social and ecological considerations that causes massive ecological degradation and the conflicting, overlapping, and inconsistent administrative and institutional structures—in order to strengthen nature conservation in China. The main objectives were to evaluate China's CPAs and the newly introduced pilot national park system (PNP) in order to: (1) identify the deficiencies of CPAs, (2) examine how the PNP model can resolve these issues, and (3) to evaluate the early successes and challenges of PNPs. The research focused on the following question: could China's pilot national park program be a solution for the ecological and administrative challenges facing CPAs. In the following sections, the term "national park" refers to these newly introduced PNPs, rather than those that function as CPAs. This research was conducted through a literature review followed by a series of workshops and discussions with national park experts working in China.

## 2. Methods

This research builds on the work published in Wang et al., 2011, as well as the outcomes of the national park project reviewed in the report on *Examining China National Park Potential and Challenges (2010–2013)* and the second phase of the Strategic Development of China National Parks in 2017–2019. The research framework and analysis process are indicated in Figure 1. Based on the research objectives, the literature review was conducted to evaluate ten of China's CPAs and the PNPs. This review follows a traditional, narrative approach by first identifying key information, then synthesizing evidence on separate topics. The gathering of academic sources, including periodicals, assessment reports, peer-reviewed literature, and government policy documents, involved keywords and acronym searches through the ProQuest Summon database. The literature review included Mandarin literature, primarily derived from the Chinese Knowledge Resource Integrated Database (CNKI)[1].

The key words used were "China" combined with "national park", "protected areas", "national park pilot", "national nature reserve", "national forest park", "key scenic and historic area", "geological park", "scenic irrigational park", "archaeological park", "forest and wildlife nature reserve", "marine park", and "cultural relics". A total of 1028 articles published since 2010 were collected using these keywords. This was reduced to 660 by using "subject", and then to 376 by using "abstract". We then used content analysis to analyze the issues presented from each selected article. The frequency of occurrence of each environmental issue in the literature was recorded and categorized based on the institutional structure of each CPA. The results were then ranked.

---

[1]     Many first-hand investigations of the study sites are only available In Chinese language, especially for those about China's pilot national parks, since the project has not yet caught enough attention world-widely. Thus, the CNKI database remains as a crucial source of knowledge for researches in China.

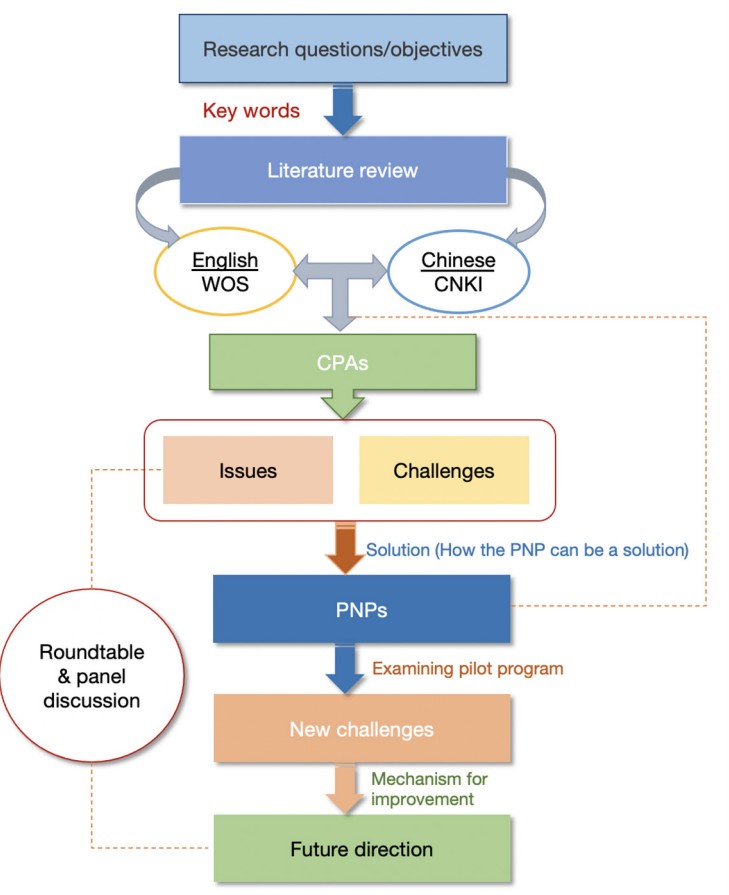

**Figure 1.** The research framework.

While the existing 10 PNPs were used for this research, 10 CPAs were selected based on the following four criteria: (1) history of establishment; (2) representativeness of key ecozones; (3) variance in types of governing bodies; and (4) level of popularity as reputed natural or cultural destinations.

The performance of China's PNPs in addressing the identified CPA deficiencies was evaluated. All available literature on park performance was reviewed and included in the pilot national park analysis. Due to the limitations of the available information on the relatively recent pilot national park initiative, additional grey literature from Chinese periodicals and news was included in our review. To ensure the relevancy of the knowledge pertaining to a quickly evolving national system of management, most of the examined literature was restricted to publications from the last 10 years. The parks selected were all designated as a national park by the pilot study. Workshops were also conducted in order to evaluate the success of the pilot national park system. These workshops were co-hosted by the University of British Columbia's National Park Research Centre and eight national park research groups in China[2].

## 3. Literature Review Findings

In this section, we examine the results of the in-depth analysis and evaluation of ten selected CPAs and ten PNPs (Figure 2), undertaken through a comprehensive literature review.

---

2 These included Peking University, Tsinghua University, Beijing Forestry University, Hainan University, Chinese Academy of Science, Chinese Academy of Forestry, Wuyi Mountain National Park, and the National Park Administration of China, respectively.

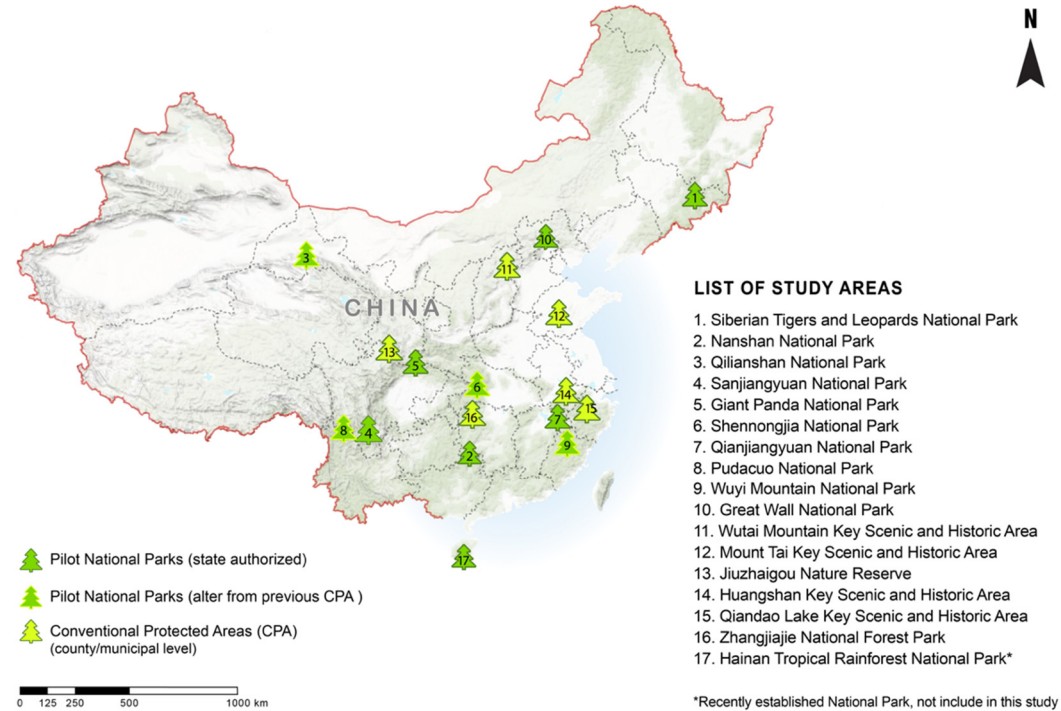

**Figure 2.** Map of study areas: the ten pilot national parks (PNPs) and the conventional protected areas (CPAs) included in the study.

### 3.1. Conventional Protected Areas

Based on the criteria listed above, we chose ten CPAs for our analysis (Table 1). These CPAs are well-established, representative of the most common categories of China's CPAs (national forest parks, national nature reserves, key scenic and historic areas, and national parks), cover three major ecozones (tropical and subtropical moist broadleaf, temperate broadleaf and mixed forest, and montane grassland and shrubland), and are managed by varying governing bodies (municipal, county, and provincial level). By reviewing literature on these ten CPAs, we determined eight outstanding environmental problems (Table 2) stemming from multiple management issues.

**Table 1.** Profile of ten conventional protected areas in China.

| Name | Est. Year | Province | Area (km$^2$) | Ecoregions | Governing Body |
|---|---|---|---|---|---|
| Wuyi Mountain Nature Reserve * | 1979 | Fujian | 565 | Tropical and Subtropical Moist Broadleaf Forest | Wuyi Mountain National Nature Reserve Administration (Provincial level) |
| Jiuzhaigou Nature Reserve | 1982 | Sichuan | 1320 | Montane Grassland and Shrubland | Jiuzhaigou National Nature Reserve Administration (County-level) |
| Huangshan Key Scenic and Historic Area * | 1982 | Anhui | 160 | Temperate Broadleaf and Mixed Forest | Huangshan Scenic Area Administration Committee (Municipal level) |
| Zhangjiajie National Forest Park * | 1982 | Hunan | 300 | Tropical and Subtropical Moist Broadleaf Forest | Wulingyuan National Forest Park Administration (Municipal level) |

**Table 1.** *Cont.*

| Name | Est. Year | Province | Area (km$^2$) | Ecoregions | Governing Body |
|---|---|---|---|---|---|
| Qiandao Lake Key Scenic and Historic Area | 1982 | Zhejiang | 567 | Temperate Broadleaf and Mixed Forest | Qiandao Lake Administration Committee (County-level) |
| Wutai Mountain Key Scenic and Historic Area * | 1982 | Shanxi | 593 | Temperate Broadleaf and Mixed Forest | Wutai Mountain Scenic Area Administration Committee (Municipal level) |
| Mount Tai Key Scenic and Historic Area * | 1985 | Shandong | 242 | Temperate Broadleaf and Mixed Forest | Mount Tai Scenic Area Administration Committee (Municipal level) |
| Shennongjia Nature Reserve * | 1986 | Hubei | 705 | Temperate Broadleaf and Mixed Forest | Shennongjia Forest Area Administration (County-level) |
| Qilian Mountain Nature Reserve | 1988 | Gansu & Qinghai | 26,531 | Montane Grassland and Shrublands | Qilian Mountain Nature Reserve Administration (County-level) |
| Pudacuo National Park | 2006 | Yunnan | 602 | Montane Grassland and Shrublands | Pudacuo-Shangri-la National Park Administration (Provincial-level) * |

* The CPAs titled with the United Nations Educational, Scientific and Cultural Organization (UNESCO) world heritage sites [23].

### 3.1.1. Environmental Issues Experienced by China's CPAs

Water Pollution: Water pollution is a major environmental problem that persists in more than half of our study sites (Table 2), with pollutants primarily caused by human activities. Four of the affected CPAs show a direct relationship with tourism facilities and activities. In Jiuzhaigou, water pollutants derived from the untreated, direct discharge sewage from local settlements has led to high mortality rates amongst aquatic species [24]. Tourist facilities in Zhangjiajie discharge wastewater after little or poor treatment, reflecting poor environmental awareness amongst local businesses and the negligence of the CPA management authorities [25]. Water eutrophication also occurs in Qiandao Lake as a consequence of unprocessed domestic wastewater [26]. Monitoring and law enforcement were found to be insufficient at Phoenix Valley located in Huangshan [27]. Other contributing factors, including mining activities and hydroelectric developments, were also found to contaminate water bodies near the study sites [28,29].

Habitat Alteration and Loss: Habitat loss was observed in six of the study sites, threatening local biodiversity and ecological integrity. A number of studies have shown that tourism infrastructure in Jiuzhaigou, such as footpaths, plank walkways, and traffic routes, fragment natural landscapes [24]. The absence of migratory corridor planning further aggravates the fragmentation of natural habitats. Excessive ecotourism also creates widespread damage to ecological corridor connectivity and alters ecosystem structure and function [30]. In Qiandao Lake, agricultural expansion, construction of infrastructure, and urbanization are the key factors threatening natural habitats [26,31]. Hydro damming projects can also alter drainage systems and water cycles, thus harming regional biodiversity [29]. Furthermore, human encroachments into natural landscapes are leading to increased instances of human–wildlife conflict [32].

Table 2. Environmental issues in the selected sites according to the literature.

| | Water Pollution | Habitat Alteration and Loss | Vegetation Loss | Soil Deterioration | Pest Outbreak and Invasive Species | Noise Pollution | Air Pollution | Climate Change | Total Number of Issues per CPA |
|---|---|---|---|---|---|---|---|---|---|
| Wuyi Mountain Nature Reserve | | √ | √ | | | | | | 2 |
| Jiuzhaigou Nature Reserve | √ | √ | | | | | | | 2 |
| Huangshan Key Scenic and Historic Area | √ | | | √ | √ | | | | 3 |
| Zhangjiajie National Forest Park | √ | √ | | √ | | | √ | | 4 |
| Qiandao Lake Key Scenic and Historic Area | √ | √ | | | √ | | √ | | 4 |
| Wutai Mountain Key Scenic and Historic Area | | | √ | | | | | | 1 |
| Mount Tai Key Scenic and Historic Area | | | | | | | | | 0 |
| Shennongjia Nature Reserve | | √ | √ | | √ | √ | | √ | 5 |
| Qilian Mountain National Park | √ | √ | √ | √ | | | | | 4 |
| Pudacuo National Park | √ | | √ | √ | | √ | | | 3 |
| Total number of Referenced Issues | 6 | 6 | 5 | 4 | 3 | 2 | 2 | 1 | |

Vegetation Loss: Vegetation degradation and loss is another leading issue affecting five of the study areas. This is largely caused by land conversion for grazing, establishment of plantations, and other agricultural production, which have effectively reduced the vegetation cover in these areas. Significant grazing in Wutai Mountain, Pudacuo, and Qilian Mountain Nature Reserve has caused degradation of alpine and subalpine grasslands [33–35]. Other human disturbances and economic developments are resulting in the conversion of natural forest to built-up areas. The harvest of Chinese White Pine (*Pinus armandii*) for building purposes is common in the Shennongjia Nature Reserve [36], and tea production in Wuyi Mountain is driving more land conversion to plantations [37].

Soil Deterioration: Four of the study sites face severe challenges associated with the decline of soil conditions and weakened fertility of the land. In Zhangjiajie, walking trails created by tourists have facilitated human incursions into previously protected natural areas [32]. These infringements not only affect the growth of the adjacent flora but compact the soil and impair the microbial balance within the surface soil. This makes the surface soil prone to erosion by natural forces such as water and wind, particularly in alpine environments [34]. Discarded waste and trampling by large mammals such as horses have increased the density of soil and reduced the porosity of the surface layer, leading to soil erosion. Furthermore, intense afforestation of coniferous trees in Qilian Mountain Nature Reserve has acidified the deep-stratum soil [35]. The rapid growth of sand breaks in CPAs reduces the extensive underground water resources, which negatively impacts the long-term fertility of the land.

Pest Outbreaks and Invasive Species: Pest outbreaks have major effects on the overall health of the ecosystem. The mixed and coniferous forests located in Shennongjia (notably *Pinus armandii*) are susceptible to outbreaks and are in critical condition [36]. The iconic Huangshan pine trees (*Pinus hwangshanensis*) are also threatened by a severe wilt disease associated with nematodes [38]. An investigation of 150 islands within Qiandao Lake found that over twenty alien species have spread across a large area, threatening endemic species and the local ecosystem [26].

Noise Pollution: Noise pollution often leads to changes in animal activity patterns and habitat range. In Pudacuo, noise is above average during the peak tourist season [34]. Excessive noise from airports disturbs local wildlife, yet the relevant authorities have paid little attention to this issue.

Climate Change: On Qilian Mountain, thawing glaciers and contracting permafrost are inducing vegetation degradation, soil erosion, and future water security, all as a result of global warming [35].

Air Pollution: Although most of the CPAs are situated far from large urban centers, a few study areas experienced high levels of air pollution. For example, waste incineration practices in Qiandao Lake emit extensive toxic gases (i.e., Dioxin, DXNs). The long lifecycles of these pollutant particles cause secondary pollution when they enter soil or water bodies. Air quality in Zhangjiajie has also declined over time as a result of growing traffic in the park [25] and regional air pollution. Most of eastern China suffers from reduced visibility associated with poor regional air quality, resulting in many of the most scenic views in CPAs being obscured.

Many of the environmental issues in the study areas can be linked, at least in part, to increased tourism. Decision-makers in CPAs often prioritize economic objectives over conservation goals, favoring tourism development. The number of "ecotourism" destinations in the country has increased from about 600 in 1990 to over 2700 in 2016 [39].

### 3.1.2. Drivers of Environmental Problems in China's CPAs

This section seeks to address and explain the issues associated with the existing CPA institutional structure and how their governance frameworks directly or indirectly contribute to these environmental challenges. These challenges are interrelated and are thus not presented hierarchically.

Complicated legislative system: Inadequate management of CPAs in China is a result of an unsound legislative system, ambiguous division of responsibilities, insufficient law enforcement, absence of central funding, and lack of professional capacity [7,14]. Aside from the 1989 Environment

Protection Law and a few other regulation codes³, legislation addressing tourism and natural resource management at the national level in China is insufficient [40]. Existing acts and policies overlook regional differences and fail to address land tenure management, finance operations, and zoning classification [41]. Without a sophisticated legal framework, a complicated and inefficient institutional structure for protected areas has evolved.

*Discordant governing systems:* Since the 1980s, the rapid development of the CPA system has led to the establishment of four ministries and three departments that branch into 13 different categories [42] (Figure 3), leading to overlapping and conflicting CPA management objectives. For instance, a 2020 study found that approximately 10% of the total PA area in Yunnan, southwest China, had been designated under multiple categories and simultaneously managed by more than one institution [43]. Although administrative agencies aim to achieve the same conservation goals, their actions are not coordinated internally [7]. These agencies do not cooperate transparently due to their distinct political focuses, leading to conflicting CPA decision-making and land management implementation. A 2019 study found that approximately half of China's 11 PA categories did not include biodiversity goals [44]. This lack of long-term land-use planning and overlapping enforcement power has led to ineffective communication between competing management authorities, exacerbating inconsistent management strategies and ineffective law enforcement [7,45]. Residents of the Phoenix village in Huangshan, for example, have complained about the lack of security guards to prevent unlawful acts by tourists on the river [27]. In Pudacuo, shortages of patrol staff have led to severe overgrazing in local meadows [7]. Jiuzhaigou experienced insufficient manpower and training to improve their emergency response team [46].

China's decentralized CPA system is further hampered by its management hierarchy, which is administered at local, provincial, and federal levels [40]. Local authorities often focus on economic development as the primary tool to increase their revenue [7]. When strict policies are enacted by higher-level governments that disregard local community needs, local authorities are reluctant to follow them [40]. Therefore, top-down regulations are not effectively implemented at the CPA-level.

Insufficient funding: The management system of China's CPAs remains ineffective due to insufficient funding. China's central government has required "local authorities to fund reserve operations" since the 1980s [7] (p. 1315). Since then, the federal government has had no further obligation to provide funding to enhance development of the CPA system. Funding, therefore, highly depends on entrepreneurial activities within CPAs, often undermining their ecological integrity [47].

Inadequate professional capacity: Training of industry professionals and public environmental education programs do not match the CPA development of infrastructure [48]. CPAs, therefore, lack adequate professional expertise and training, resulting in undertrained staff and park managers that stress investment in physical capital over investment in human capital. Some studies estimate that "only about a third of China's Nature Reserves' employees have adequate training", while the rest remain unsuitable for their tasks [7]. It is worth noting that most nature reserves are situated in remote areas, where employment opportunities are not attractive to recent graduates with better training.

Land-use dilemma after China's forest tenure reform: In 2003, an ambitious tenure reform was undertaken in China to transfer collective-owned forest land into the ownership of private households to better secure their income [49]. However, much of the collective-owned forest land was protected land within CPAs. The government enacted legal documents, such as the Forest Acts and Regulations on Nature Reserves, imposing tight controls on the operational activities inside CPAs [50]. Although farmers obtained rights to their forestland, they remained under strict rules, and profits from the land have been curtailed [49]. This inconsistency in the "de-collectivization" land tenure policy both limits the capacity to manage protected land sustainably and undermines trust in the local authorities.

---

³ The Nature Reserve Regulation of P.R.C. (1994); The P.R.C. Scenic Area Regulation (2006).

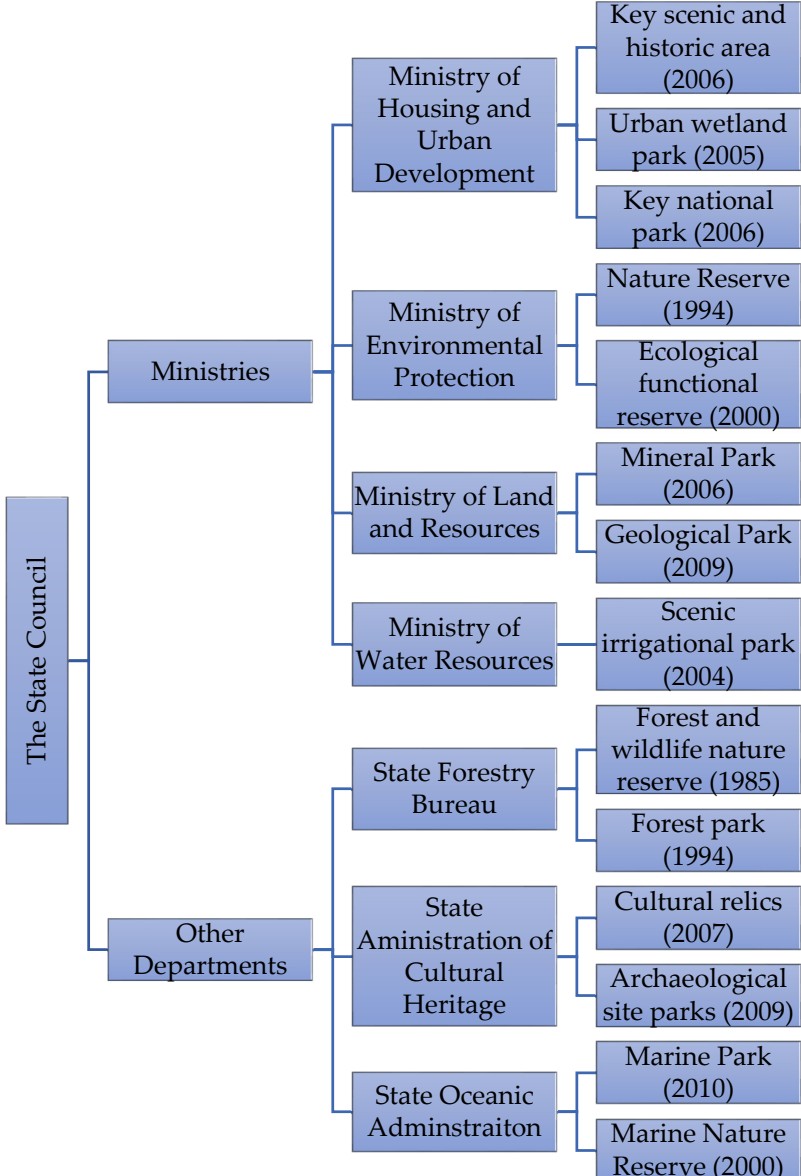

**Figure 3.** The conventional protected area (CPA) institution structure in China before 2018 (adapted from [40,41]).

Overexploitation of natural resources: In China, legal documents such as the Mineral Resources Act 2009, the Nature Reserve Regulation 2011, and the KSHA Regulation 2006 have been established to prohibit unauthorized resource extraction activities in protected areas [51–54]. However, hydroelectricity and mining operations in CPAs are still being conducted, despite existing laws [55]. A 2017 study estimated that there are 104 quarries, 318 industrial mines, and 335 energy facilities present in China's Nature Reserves [56]. One report suggested that 580 illegal mining and quarry operations were active in 86 national nature reserves [57].

Many large hydropower dams, such as those on the Nujiang, the Da Du, and the Minjiang Rivers, are situated in ecologically sensitive areas [58], resulting in further ecological threats. Qilian Mountain CPA contains a wealth of water resources for hydroelectricity; currently, there are 42 power stations located within the protected area [59]. Design defects of these power facilities along with a lack of environmental assessment prior to construction have resulted in downstream water shortages, drainage basin degradation, and regional biodiversity losses.

The Ministry of Ecology and Environment and the State Council have both called for the withdrawal of all mining and quarry companies from nature reserves in 2015 and 2016, respectively. In 2017, the Ministry of Natural Resources announced they would no longer grant mining rights in any reserves [51]. Although some ecological compensation plans have been proposed, detailed procedures remain unconfirmed and have delayed efforts to undertake environmental restoration.

Overall, the current CPA regulatory system lacks environmental legislation and suffers from unclear land and administrative demarcation, discordance between governing units at all levels, ineffective law enforcement, a lack of reliable funding, a shortage of nature conservation professionals, insufficient environmental education, and overlapping responsibilities among three tiers of government. Together, these challenges have led to poor management efficiency and effectiveness. These problems have been recognized by the central government, and the policy response has been to introduce a pilot program for a national parks system in China.

### 3.2. Pilot National Parks

Since 2015, China has been developing a national park system, which includes 10 pilot national parks in 12 provinces across China, covering more than 200,000 km$^2$ of land [21]. These national parks aim to protect the most important flagship species, such as the Giant Panda (*Ailuropoda melanoleuca*), the Siberian Tiger (*Panthera tigris* ssp. *altaica*), the Asian Elephant (*Elephas maximus*), and the Snow Leopard (*Panthera uncia*), as well as key natural and cultural heritage sites in China. This section provides an overview and evaluation of the ten PNPs.

As of 2018, all ten PNP designations have been confirmed. Some basic information for each PNP is given in Table 3. On 1 April 2019, the eleventh pilot national park, Hainan Tropical Rainforest National Park, was formally established [60], but it was omitted from our analysis as administrative information was not available at the time of data collection.

**Table 3.** Profile of the original 10 pilot national parks in China.

| Name | Date of Establishment | Province/City | Area/km$^2$ |
|---|---|---|---|
| Pudacuo National Park | 2015 | Yunnan | 602 |
| Qianjiangyuan National Park | June 2016 | Zhejiang | 252 |
| Wuyi Mountain National Park | June 2016 | Fujian | 983 |
| Shennongjia National Park | May 2016 | Hubei | 1170 |
| Nanshan National Park | July 2016 | Hunan | 619 |
| Siberian Tigers and Leopards National Park | December 2016 | Jilin, Heilongjiang | 14,612 |
| Sanjiangyuan National Park | January 2018 | Qinghai | 123,100 |
| Great Wall National Park * | November 2017 | Beijing | 60 |
| Giant Panda National Park | April 2017 | Sichuan, Gansu, Shanxi | 20,200 |
| Qilianshan National Park | June 2017 | Gansu, Qinghai | 52,000 |

* The Great Wall National Park may leave this pilot program due to its complexity in nature, culture, and local politics. However, no official announcement has been made. (Sources from [18,19,22,61]).

The spatial allocation of the ten PNPs was determined based on the boundaries of existing CPAs as well as each area's representativeness for multiple identified ecological values. These factors are established in Table 4.

**Table 4.** Profile of ten pilot national parks in China.

| Name | Overlap with Existing CPAs | Protection Value Significance |
| --- | --- | --- |
| Pudacuo National Park | No overlap with existing CPAs but is home to endangered species, geological, wetland, water, and forest landforms | • Natural values of geological, wetland, water, and forest landforms<br>• Habitat/biodiversity protection and restoration<br>• Representative of the plateau ecosystem and alpine lacustrine environment in southwestern China |
| Qianjiangyuan National Park | • Qianjiangyuan National Forest Park<br>• Gutianshan Nature Reserve<br>• Qianjiangyuan Tourist Attraction | • Natural values of water, wetland, and forest resources<br>• Habitat/biodiversity protection and restoration |
| Wuyi Mountain National Park | • Wuyi Mountain Nature Reserve<br>• Wuyi Mountain Key Scenic and Heritage Area<br>• Wuyi Mountain National Forest Park | • Natural value of forest resources<br>• Biodiversity protection for over 7500 flora and fauna species (key biodiversity hotspot globally)<br>• Representative of the subtropical forest ecosystem in Southeastern China |
| Shennongjia National Park | • Shenongjia National Forest Park<br>• Shenongjia Nature Reserve<br>• Shennongjia Geopark<br>• Shenongjia Wetland Park | • Natural values of geological landforms and forest, water, and wetland resources<br>• Habitat/biodiversity protection and restoration (particularly the Snub-nosed Monkey) |
| Nanshan National Park | • Nanshan Tourist Attraction<br>• Jintongshan Nature Reserve<br>• Baiyunhu National Wetland Park<br>• Liangjiangxiagu National Forest Park | • Habitat/biodiversity protection and restoration<br>• Natural value of water resources<br>• (Particularly the Yangtze River and the Pearl River Basin)<br>• Representative of the subtropical low-altitude evergreen eco-community |
| Siberian Tigers and Leopards National Park | No overlap with existing CPAs but covers abundant temperate flora and fauna across two provinces | • Natural value of forest resources<br>• Habitat/biodiversity protection and restoration (particularly Amur Tigers and Leopards) |
| Qilianshan National Park | No overlap with existing CPAs but covers surrounding ecosystems including rivers, forests, glaciers, and desert. | • Natural values of water, forest, glacier, and desert resources<br>• Habitat/biodiversity protection and restoration (particularly Snow Leopards, Red Deer, and antelope)<br>• Representative of the natural environment in Northwestern China |

**Table 4.** *Cont.*

| Name | Overlap with Existing CPAs | Protection Value Significance |
| --- | --- | --- |
| Great Wall National Park | • Badaling Great Wall Heritage<br>• Yanqing Geopark<br>• Badaling Forest Park<br>• Ming Tombs | • Cultural/historical value for Chinese civilization<br>• The Great Wall is one of the symbols of Chinese civilization<br>• The Ming Tombs are one of the symbols of Chinese history<br>• Natural values of geological landforms and forest resources |
| Giant Panda National Park | • Chengdu Research Base of Giant Panda<br>• Existing Mountain Systems of Minshan, Daxiangling, Qinling, Baishuijiang and Ganan. | • Cultural/historical value for Chinese civilization (Giant Panda is one of the symbols of China)<br>• Habitat/biodiversity protection and restoration (particularly the Giant Panda)<br>• More than 8000 wild flora and fauna |
| Sanjiangyuan National Park | • Kekexili National Nature Reserve<br>• Headstreams of Yangtze River, the Yellow River, and the Lantsang River | • Natural value of water resources (Particularly the Yangtze River, Yellow River, and Lantsang River)<br>• Habitat/biodiversity protection and restoration |

(Sources from [18,19,35,36,59,62–71]).

### 3.2.1. Key Guiding Principles and Reform Mechanisms

The establishment of China's PNP initiative is based on the practical experience of CPAs and the challenges described above. In order to be included in the pilot national park system, CPAs were required to demonstrate that they had the following key guiding principles:

Ecological/cultural representativeness: Ecological and cultural values are important considerations in identifying nationally significant priority conservation areas [72–74]. Currently, more than 50% of the CPAs are concentrated in the central and the eastern regions of the country, despite the high natural resource values present in the western region [75]. In designing the PNP geographic scope, experts recommended CPA identification based on this representativeness, as there is no natural ecosystem in central or eastern China that is large enough to meet the area criterion of a national park [72].

State dominance (top-down design): A centralized administrative approach to natural landscape management is key to addressing management inconsistencies resulting from numerous overlapping and conflicting governing units of the current system. There has been widespread support for centralizing the administration of prospective PNPs [18,76,77]. In this way, park systems can enforce the important principles of uniformity, standardization, and high efficiency [78].

De-commercialization: Conservation is the primary goal of national parks. Shifting the focus from profitable economic activities that conflict with conservation objectives in parks to protection and restoration efforts has been widely emphasized [72–74]. It has been suggested that operations that neither consume resources nor damage the environment should be considered. Ecotourism, for instance, has played an important role in the economic and the social development of the Giant Panda National Park, but its persistence will likely depend on better enforcement of the relevant legislation to ensure the park's ecological integrity [79]. It has been recommended that the profits from the remaining economic activities be used to serve the surrounding communities and to enhance future protection and management of the national park [72].

Public participation/engagement: PNPs are seen as welfare goods that provide recreational and educational benefits to the public at low cost (or even for free) [79]. Ecological education and other related efforts have therefore been highlighted as key approaches to promote public engagement and participation in park conservation [18,77,78,80].

Science-based management: Science-based decision-making is the key to identifying priority areas and to determine park management strategies. Experts recommend identifying priority areas for PNP inclusion through careful evaluation of science-based criteria and indicators. One study has already evaluated protection values of China's natural landscapes and identified over 8% of the country's landmass as priority areas for park planning [81], and another noted a 19% spatial mismatch of priority areas with current nature reserve locations [82]. It has been emphasized that prospective national parks should establish scientific research bases for further environmental or ecological studies that contribute to their sustainability [72].

Ecosystem integrity: Restoration and maintenance of ecosystem integrity are underscored in discussions on PNP management plans [74]. Experts suggest establishing the boundary of each national park on the basis of integrated and coherent ecosystems such as mountain chains, hydrographic basins, or habitat type as a geographical unit for demarcation, as opposed to administrative boundaries [72]. In this way, the core resource or significance of each national park can be more effectively protected.

Given these six guiding principles of PNPs, the development of China's pilot national park system currently focuses on the following five reform mechanisms.

Prioritization of ecological protection: Given their emphasis on ecological integrity, PNPs are focusing on resource assessment (natural or cultural) by classifying levels of degradation and identifying best management practices [18]. PNPs are also re-assessing activities and facilities that could negatively impact local ecosystems and are reorganizing land boundaries. This is being approached through an introduced four-tier zoning system that distinguishes between strict protection zone (SPZ), ecosystem conservation zone (ECZ), native community zone (NCZ), and research education and recreation zone (RERZ) [18,83]. This has already been determined to be effective in some parks [84]. Experts further

suggested specifying a framework for the pilot national park system under an ecosystem services approach to achieve conservation objectives while promoting their social benefits [85,86]. Other studies suggest adopting a biosphere reserve approach to accommodate anticipated high human activities while preserving core protected areas [87].

Unification of management standards: Fragmented or overlapping management systems have been a key challenge in China's CPAs, resulting in poor ecosystem protection and environmental degradation. The unification of management standards through a top-down design implemented by a centralized approach can help resolve this issue [18]. The aim is for each national park to be reorganized into vertical management institutions under a single governing body, the National Park Administration (Ministry of Natural Resources), under the State Council [18].

Clarification of resource ownership: As centralized governance and public interests are two important guiding principles of a national park, studies emphasize that access to these natural landscapes should be publicly available and state-owned [18,88]. It has been suggested that each PNP should conduct an initial evaluation of any active or potential resources that can be extracted and then legally transfer them to the state via lease, levy, legal agreement, stock partnership, or asset purchase [18].

Innovation of operating management: Many studies encourage prospective PNPs to modify their business model to ensure they are operating in the public's interest. Suggested approaches include implementing low-cost or free entrance, thereby encouraging public access [18,88]. Furthermore, public-private partnerships or private operations based on a resource allocation that follows market principles have been highlighted as a useful approach to reduce operating costs [18]. Studies emphasize the formulation of regulations to ensure that the permitted businesses and tourism are not environmentally harmful [13,80].

Promotion of community development: PNP success depends on the quality of the park's relationship with the surrounding communities [18]. It has been estimated, for instance, that about 5500 local residents live in the core protection zone of the proposed Giant Panda National Park [89]. If the cancellation of projects and facilities or restrictions on the use of land and resources in the park impacts nearby residents, their employment and compensation should be taken into consideration. A good relationship with local stakeholders encourages the public to participate in the planning and the management of the park [18]. Where needed, education and training on ecological significance should also be developed to reinforce awareness of environmental sustainability and to further promote community participation.

3.2.2. Evaluation of China's Pilot National Parks in Relation to IUCN's Admittance Criteria for National Parks

The key indicators that determine whether an area can be officially categorized as a national park by the IUCN include area, resource class, human footprint index, and functional complexity [77]. According to the IUCN, an area under consideration for national park status must encompass more than 10 km$^2$, be nationally representative in terms of its resources, have a minimal human footprint index, and promote complex ecological functions (including scientific, educational, and recreational uses).

All ten PNPs in China meet the area requirement of IUCN. Under the resource class criteria, IUCN focuses on national representativeness. Although the formation and the operation of China's PNPs occur at the national level, some of their resource classes remain at the provincial or the municipal level [77]. The human footprint index weighs human influence on the ecosystem of the national park based on "population pressure", "land utilization", and "infrastructure construction" [77]. All of China's PNPs fare poorly with regard to this criterion due to the high influence of humans on their ecosystems. Table 5 is an adaptation of Tian and Fang's 2017 findings of indicators analyzed in eight PNPs in China and their capacity to meet IUCN's requirements.

**Table 5.** Comprehensive index of eight pilot national parks in China.

|  | Great Wall | Qianjiang Yuan | Wuyi Shan | Shen Nongjia | Nan Shan | Pudacuo | Sanjiang Yuan | Siberian Tigers and Leopards |
|---|---|---|---|---|---|---|---|---|
| Area | 0.21 | 1.78 | 3.99 | 8.26 | 4.37 | 9.27 | 43.45 | 13.47 |
| Resources Richness | 11.63 | 10.47 | 11.63 | 11.63 | 11.63 | 10.47 | 11.63 | 11.63 |
| Human Footprint | −17.35 | −16.67 | −18.03 | −7.14 | −9.52 | −17.69 | −3.06 | −4.42 |
| Function complexity | 2.79 | 5.18 | 9.47 | 13.63 | 8.98 | 12.01 | 14.12 | 15.78 |
| Overall Score | 0.42 | 2.07 | 4.62 | 9.55 | 5.42 | 9.28 | 38.46 | 14.34 |

(Source from [77]).

Sanjiangyuan National Park is the study area that fits most closely with IUCN's definition of a national park, whereas the Great Wall National Park fails to meet the IUCN guidelines (Table 5) [77]. Of all the criteria, it is clear that human footprint is the most significant challenge for China's PNPs. The human footprint is aggravated by common park practices that include transportation, mass tourism development, poor water treatment, and solid waste accumulation [47]. A heavy human footprint is an inevitable issue in China, a country with one of the largest populations that has prioritized economic development for decades [77].

While mitigating the human footprint in PNPs poses a significant challenge, the other criteria are being advanced with some success. Regarding park area, for instance, Wuyi Mountain, Shenongjia, and Qilianshan have each almost doubled in size [90]. The Giant Panda National Park aims to consolidate 81 individual protected areas to create a cohesive management structure [91]. These efforts indicate China's dedication to meeting internationally agreed-upon ecological standards of protected areas.

## 4. Future Considerations in China's Pilot National Parks

As discussed above, the implementation of a national park system in China is a helpful way to address a broad range of administrative, legislative, and social challenges faced by CPAs. Under this system, the roles and the responsibilities of various departments are clarified through the introduction of a centralized National Park Administration, efforts in law-making, as well as the amendment of existing policies that alleviate the statutory vacuum that characterized the previous system.

Figure 4 presents the institutional structure of the PNP system. Under the State Council, the National Forestry and Grassland Administration, which is also officially named as the National Park Administration, is responsible for all PNPs [20]. Three of them, the Siberian Tigers and Leopards National Park, the Giant Panda National Park, and the Qilianshan National Park, are directly managed by the National Park Administration, as they are cross provincial boundaries. The remaining seven PNPs are co-managed by the centralized National Park Administration and the provincial governments where each PNP is located. The authority of the co-management is clear and divided such that the National Park Administration is responsible for ecological protection, natural resource management, public engagement, public education, and scientific research, while the provincial governments are responsible for the in-park economic development, local community management, public services, natural disaster prevention, and market supervision [20]. Unlike the institutional structure of the CPAs (Figure 3), with the involvement of multiple ministries, issues such as management overlaps and inconsistent mandates can be mitigated in the PNP model due to the vertical administration structure.

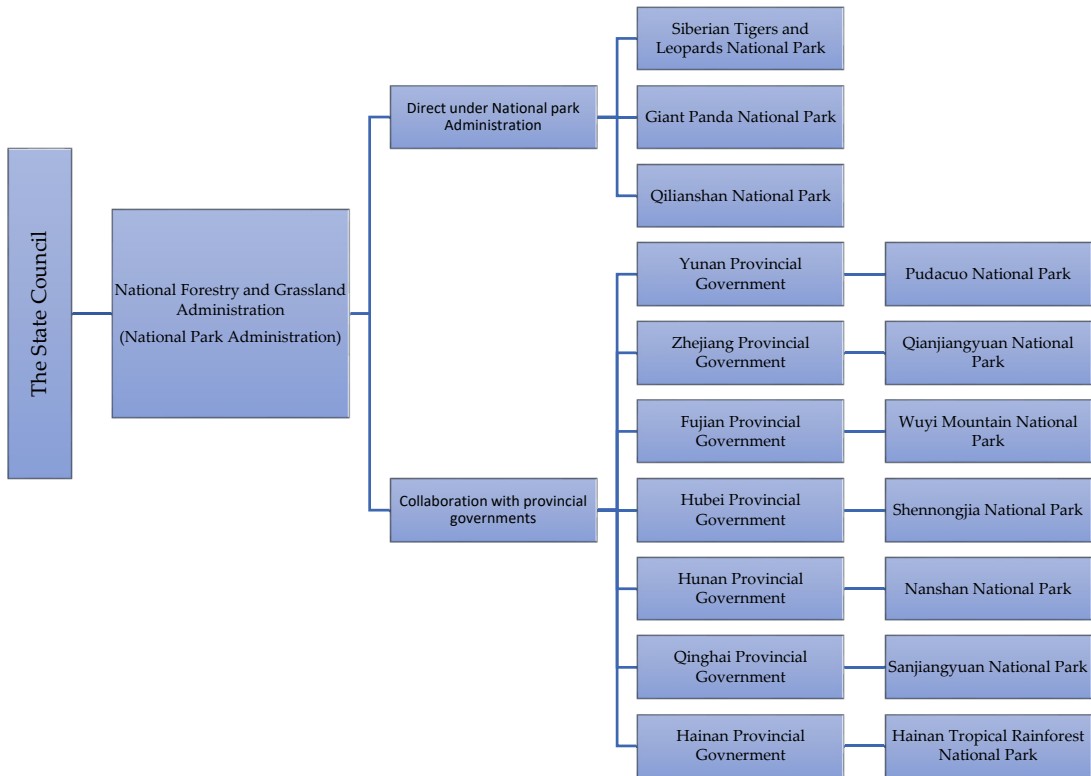

**Figure 4.** The pilot national park (PNP) institution structure in China (source from [20]).

Five national parks (Sanjiangyuan, Wuyi Mountain, Shennongjia, Nanshan, and Qianjiangyuan) have already begun to adopt policies to shift administration to the more centralized system, establishing a "one national park, same title, single authority" governing structure [19]. Already, there is evidence that these new parks are meeting their objectives. For instance, Sanjiangyuan National Park Authority has formulated several administrative measures that incorporate aspects of ecological management and budget control, alleviating community poverty by creating 7421 jobs while simultaneously pursuing environmental protection objectives [92].

However, the introduction of the PNP initiative has been criticized for not explicitly addressing reforms to the CPA system and for an apparent lack of centralized leadership [93]. The following issues have been identified by professionals (e.g., [19,79,81]) as key considerations to future national park management:

Land ownership: Much of the land in the PNPs was once owned by collective villagers, local government, or provincial government. The extension of the park regions under the new system along with the transfer of land ownership to the central government implies economic losses for many farmers and local governments [13]. Therefore, the biggest challenge to making this land transition successful is providing compensation to the farmers or the local governments for their losses. Qianjiangyuan National Park pilot has already had some success in its transition, as most local residents reported no change in their net annual income (53%), and some (37%) reported a net increase [94]. That said, the compensation should be based on their income level prior to the PNP establishment to ensure that the local residents and the governments do not experience economic losses due to the land transition. Scholars have concluded that the issues of land transition or ownership should be resolved by respecting the will of local residents, strictly specifying and managing the process of land acquisition, increasing the standard of compensation, and balancing the goal ecological protection and the provision of benefits to local residents/farmers [20].

Personnel: The transformation of China's CPAs selected to become national parks implies a shift in management from the regional/provincial level to the national level. Likewise, the management of

human resources must also be transitioned into the centralized system. As a result, park staff whose expertise of park management was at the regional/local level might be struggling in their national positions. This personnel shift has yet to be organized in a way to best facilitate skill transitions. We recommend that a personnel re-training program be instituted to help regional/provincial park staffs accommodate their new centralized management positions at the national level.

Economic re-zoning: Re-zoning economic activities in order to facilitate better conservation efforts while ensuring local livelihoods could prove to be a challenge. It has been estimated, for example, that 11.7% of panda habitat within existing protected areas of the Sichuan portion of the Giant Panda National Park has no restrictions on either timber extraction or human disturbance [95]. However, this results in significant changes to the distribution of economic benefits for those people who used to work inside park boundaries. Experts suggested offering alternative employment in park management or tourism. Furthermore, some studies have suggested that it is possible for concessions to remain within national parks as long as they are guided by a concessions program that prioritizes natural resources as a public good [96]. Although there may be other opportunities, such as tourist transportation or accommodation in local communities, the economic redistribution of these opportunities is a challenge.

Traditional knowledge: A significant debate has emerged over whether to resettle local communities living inside the park (risking human rights violations) or allow them to remain (risking an adverse human footprint). This debate suggests that the PNP system is not an inclusive bottom-up mechanism [97]. Furthermore, removing local communities risks the loss of traditional ecological knowledge (TEK), which is very important for the effective management of protected areas [98]. Balancing the contributions of these communities with their human footprint in the parks is a necessary element to engaging local park support. To address this, the PNP authorities should carefully design the native community zone (NCZ) of each PNP so that local communities are incentivized to relocate there voluntarily, embracing the principles of informed consent, participation, and sufficiency to maintain their quality of life [99]. Further, local stakeholders (especially TEK holders) should join the decision-making table with scientists and policymakers for the ecological management of the PNP system [20]. The Shennongjia and the Giant Panda National Parks have already employed local residents as ecological guardians in the parks [20].

Financial management system: It is expected that the national government takes financial responsibility as PNPs transition from a local to a more centralized system. Although this transition frees provincial governments from the financial burden of funding the parks, it also deprives provincial governments of some profits via tax revenues. However, activities such as commercial re-zoning or the application of TEK must rely on local stakeholders. As a result, provincial or municipal governments may remain financially responsible for park operations. Thus, the revenue distribution and partnership between the central and the provincial governments must be managed carefully. A financial design managing the disbursement of funding from the centralized National Park Administration to local actors for PNP operations and management seems to be a fair solution

Law enforcement: A lack of effective law enforcement to ensure environmental protection has been a challenging issue for China's CPAs, resulting in significant environmental degradation. Most environmental regulations and prohibitions exist in name only, since law enforcement is limited to warning individuals and companies that violate rules without *de facto* punishments. Although the administration of PNPs has been centralized at the national level with an increased focus on environmental protection, approaching improved law enforcement in these regions remains a challenge. The PNP authorities should begin designing and codifying the *de facto* punishments into a statutory law system.

## 5. Conclusions

The reforms identified here, aimed at helping the establishment of national parks in China, could inform other countries whose protected areas have faced similar administrative challenges or environmental pressures. For instance, not unlike China, Indonesia is experiencing protected area

erosion due to infrastructure development and discordant governance [100]. Indonesian scholars can draw lessons from China's experiences and avoid the mistakes made during China's transition to a national park system.

Still in its early stage of development, China's pilot national park system remains disorganized in practice and is not currently able to address the challenges articulated in this study effectively. Possible solutions to perfect this model should integrate lessons learned from other countries with well-developed national park systems. For now, we strongly suggest that the central authorities in China prioritize the development of a set of laws and regulations applicable to national parks. Canada's national park legal system provides an appropriate model, as its comprehensive nature encompasses by-laws including but not limited to wildlife, endangered species, hunting, fireproofing, traffic, architecture, and community involvement. [101]. Future work on China's national park system could focus on adaptation to current trends and challenges by adopting some of the best practices developed in other parts of the world. Furthermore, as the pilot national park system is still in its early stages, an evaluation of its success might be premature. A follow up study should examine the system's effectiveness after the final stages of implementation.

**Author Contributions:** G.W.: Conceptualization, Methodology, Writing—Review & Editing. J.L.I.: Conceptualization, Review & Editing. G.S.: Resources, Data Curation, Writing—Original Draft. H.C.: Resources, Data Curation, Writing—Original Draft. Y.Z.: Resources, Data Curation. K.F.-G.: Resources, Data Curation, Writing—Original Draft, Writing—Review & Editing. Z.W.: Writing—Original Draft. All authors have read and agreed to the published version of the manuscript.

**Funding:** This research was supported by the UBC Global Seminar Program and the one-month Go Global Study tour to China, and the APFNet Research Project (APFNet 2017 Sp2-UBC).

**Acknowledgments:** We would like to thank Jun Yang of Tsinghua University, Dihua Li of Peking University, Jian Wu of Renmin University of China, Wenqiang Xu of the Chinese Academy of Science, Xiqiang Song of the National Park Research Institute at Hainan University, Wenzhou Cheng of Wuyi Mountain National Park, Zhaogang Duan of the Giant Panda National Park, Lvbei Yi of Sanjiangyuan National Park, and Guangcui Dai of the National Park Administration of China for co-hosting research workshops and roundtable discussions and, more importantly, for their insights into national park and protected areas management in China.

**Conflicts of Interest:** The authors declare that they have no conflict of interest.

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
