# Peer review of "Moving toward a Greener China: Is China’s National Park Pilot Program a Solution?"

_land, doi:10.3390/land9120489_

Round 1

Reviewer 1 Report

The paper is quite well written and provide as good overview of the current state of the management of China’s incipient  National Park System. The paper is an interesting and informative. It has, however a number of shortcomings that need to be addressed before the paper can be published.

The Abstract claims that the “The study highlights a series of potential solutions to address these identified challenges”. The paper does not do this, however. The challenges are identified and described, but the solutions are not advanced in any level of detail. This really needs addressing.

The conclusions section is not very well developed. Section 4 sets out the challenges, but does not address the concluding take home message. What is the take-home message for anyone in the Chinese administrative and decision-making system?   What are possible solutions, if any can be advanced?

The formulation used in line 476  f (‘can be used by other countries whose protected areas are less developed.”) can be read as exceedingly arrogant. I would counsel to tone this down to something like “can be used to inform other countries whose protected areas face administrative challenges or economic development pressures.”

The concept of “Transboundary Protected Areas” is thrown in at the end (line 481), with a geopolitical ambit “bilateral or multilateral cooperation with neighboring countries”. This is totally unconnected from the text and does not really flow from the previous sections. It should either be dropped, or the idea has be well developed…but then falls outside the focus of the paper as it stands. I recommend to drop and develop that idea as a separate paper.

Section 4 (Future Challenges in China’s Pilot National Parks) does not address the issue of administrative “territoriality.” As Figure 3 shows, there is a plethora of agencies that control Conventional Protected Areas. Clearly Figure 3 is evidence of political fiefdoms that exercise command-and-control over resources and assets. It is likey, however, that all Ministries have the same political weight…What is the relationship of the newly founded  ‘centralized National Park Service Bureau’ to the Ministries and other agencies shown in Fig 3? Does the NPSB stand a chance or will it be undermined in its independence (if it has any) or subsumed by other ministries protecting assets under their control? This issue needs to addressed or at least discussed and acknowledged, even if ultimately the authors may not be able to advance a solution and may have to ‘park’ the issue.

Detailed comments

The authors may want to peruse and incorporate the following papers in their intro section:

Fan, Yangcheng. "Field Philosophy and the Chinese National Park System." Social Epistemology (2020): 1-11.

Wang, Ju-Han Zoe. "National parks in China: Parks for people or for the nation?." Land Use Policy 81 (2019): 825-833.

Huang, Qiongyu, Yuxiang Fei, Hongbo Yang, Xiaodong Gu, and Melissa Songer. "Giant Panda National Park, a step towards streamlining protected areas and cohesive conservation management in China." Global Ecology and Conservation 22 (2020): e00947.

Wei, Dongying, Aixia Feng, and Jingyi Huang. "Analysis of ecological protection effect based on functional zoning and spatial management and control." International Journal of Geoheritage and Parks 8, no. 3 (2020): 166-172.

Wang, Weiye, Jinlong Liu, and John L. Innes. "Conservation equity for local communities in the process of tourism development in protected areas: A study of Jiuzhaigou Biosphere Reserve, China." World Development 124 (2019): 104637.

Line 53 ff It would be sensible to briefly include 2-3 sentences with references to comment on any parks/conservation areas/reserves used (if any) pre World War II China, ie during the Qing Dynasty as well the Republic of China period. This would set up the historic trajectory. Some form of conservation management surely must have existed…

Line 59 “Conservation equity for local communities in the process of tourism development”  (ref above)  states 2700 PAs…:

Line 75 suggest to beef up statement with more references

Line 78, “resulting in degradation of the country’s local ecosystems”  needs reference to support the claim/statement.

Lines 85-86: “to resolve the outstanding problems for nature conservation in China.” The outstanding problems have not yet been well defined…this needs beefing up in the preceding paragraph

Line 99-100 “involved acronyms and 99 keywords searches” give the search logic, please

Line 104: “A qualitative analysis was conducted” which methodology was used?

Line 112: “designated a national park by the project.” Which project? Explain/make clear

Line 114f: “A qualitative analysis was 114 used to determine” which methodology was used?

Line 120: Make the image bigger (full widh) hard to read in page view as it stands

Line 131: Include the category of the CPA in the table

Line 133: Figure. Typo “Habitats…should be Habitat

Line 133: I would make that a cross table CPA vs issue and then use a tickmark were it applies, with total counted frequencies per row and column. The image does not tell me that much

Line 136: 3.1.1. heading is incomplete

Line 148: “Habitat loss was observed in five of the study sites,” Figure 2 says six…

Line 163 “Pinus armandii” needs to be in italics

Line 247: Typo ‘Chin”

Line 255f:  Cite the formal titles and years of the acts (as Endnote..)

Line 282: Species need to be italics…check this through the document, will not comment on this from hereon..

Line 287: check Grammar

Line 289. I would move the footnote 4 to the end of line 287

Line 306: needs reference to support the claim/statement.

Line 392: something went wrong with the citation

Line 455: something went wrong with the citation

Formatting commnets:

The various issues, shewn on pages 6ff are one issue per paragraph. The readability would benefit greatly  if each issue at the start of the paragraph were italicised

I have not commented on table layout and formatting as that will be, I presume, addressed at the production stage..

Author Response

Dear Reviewer I.

Thank you for giving us the opportunity to submit a revised draft of my manuscript. We greatly appreciate the time and effort that you and the reviewers have dedicated to providing your valuable feedback on our manuscript. We are deeply grateful to you for your insightful comments on our paper. We have been able to incorporate changes to reflect most of the suggestions provided by you all.

Attached please find a point-by-point response to your comments and concerns.

Again, many thanks for your excellent comments on our paper!

Best wishes,

Dr. Guangyu Wang, on behalf of the co-authors

Reviewer 2 Report

Comments and Suggestions for Authors

The manuscript is a review of China's protected area system with a special focus on the development of conventional protected areas and the pilot national parks. The study highlights the main environmental and administrative problems of protected areas and offers possible solutions. The manuscript is interesting, well-written, and worthy of publication, but requires extensive editing in order for it to be published. The reviewer strongly recommends authors to re-examine the methods and results sections as the data used in the review are unspecified, the analysis is not clear, and the results are not clearly represented. This will drastically affect the understanding and quality of the study.

Major Issues

Title

The title sounds very interesting, but the work is mainly focused on conventional protected areas (CPAs).

Introduction

The introduction is very well structured, but the aim and research questions of the study are not mentioned in the introduction. To clearly understand the purpose of the paper, please add it. I suggest moving the first paragraph of methods to the introduction.

Line 35-36: A citation and some examples can be given.

Methods

In the methods section, the criteria for choosing CPAs and the approach used are clearly presented. Although a keyword search has been used in the literature and the ProQuest Summon database, no mention is made of the criteria for article screening and keywords used. The total number of resources analyzed and the final number of sources considered are not given. What was analyzed, the title, abstract, or the full article? It is mentioned that the results were analyzed quantitatively, but the method is not mentioned.

It was mentioned that the literature analyzed was mainly published in the last 10 years. A diagram with all the literature analyzed should be presented. Each phase should be documented for reasons of transparency and reproducibility.

The results are shown in Figure 2. However, this is far from sufficient. I advise authors to do a network diagram analysis and display the results as a network diagram illustration that will help readers better understand. The authors mentioned that the CPAs analyzed are affected by certain environmental problems. However, are they equally affected? Can you quantify this?

Are there methodological and data-related limitations? If so, they should be mentioned.

Lines 98-100 and 106-115. The number of sources used and the exact source should be specified on the support materials. What method did you use to extract the information?

Line 121. Figure 1. Please consider changing the size of the legend and the labels as they are not visible. For better visibility, the map can be focused on China or the exact regions that are being analyzed, with the exception of regions in the far north, which are not of interest to this study.

Line 131. Table 1. The font size should be smaller for all tables

Figure 2. The x-axis is not clear. Please consider improving the quality and correcting the spelling mistake "noise pollution". Table 1 lists 10 conventional protected areas, but the relationship between CPAs and environmental issues is not clear in this graph.

Please make a clear connection between the CPAs studied and the environmental issues. Also, consider using a quantitative analysis that will help you highlight the number of environmental issues and CPAs.

Lines 137-138. “more than half of our study sites” – I suggest writing the study site numbers/names. The same in lines 148, 158,166, and the other sections.

Results

The environmental problems identified are very well explained, but no clear link is made between these problems and the CAPs studied.

The proposed solutions to address the identified challenges are well presented in the text but they should be prioritized or rated in such a way that their benefits can be quantitatively highlighted for each protected area. This would be helpful for the reader to understand what is the most efficient approach.

The challenges that CPAs and national parks face are very well discussed in the text. However, some of them overlap. Therefore, I recommend implementing an approach that can identify common and unique problems and show them through a graph.

Conclusion

The conclusions section contains useful recommendations for other countries.

Author Response

Dear Reviewer II.

Thank you for giving us the opportunity to submit a revised draft of my manuscript. We greatly appreciate the time and effort that you and the reviewers have dedicated to providing your valuable feedback on our manuscript. We are deeply grateful to you for your insightful comments on our paper. We have been able to incorporate changes to reflect most of the suggestions provided by you all. Attached is a point-by-point response to your comments and concerns.

Again, many thanks for your excellent comments on our paper!

Best wishes,

Dr. Guangyu Wang, on behalf of the co-authors

Round 2

Reviewer 1 Report

Please commend the authors on their serious approach to my review comments and their comprehensive response.